# MAE-Based Self-Supervised Pretraining Algorithm for Heart Rate Estimation of Radar Signals

**DOI:** 10.3390/s23187869

**Published:** 2023-09-13

**Authors:** Yashan Xiang, Jian Guo, Ming Chen, Zheyu Wang, Chong Han

**Affiliations:** 1School of Computer Science, Nanjing University of Posts and Telecommunications, Nanjing 210023, China; b20032124@njupt.edu.cn (Y.X.); b21030308@njupt.edu.cn (M.C.); b21030123@njupt.edu.cn (Z.W.); hc@njupt.edu.cn (C.H.); 2Jiangsu High Technology Research Key Laboratory for Wireless Sensor Networks, Nanjing University of Posts and Telecommunications, Nanjing 210023, China

**Keywords:** FMCW radar, heart rate estimation, self-supervised pretraining, transfer learning, masked autoencoders, vision transformer

## Abstract

Noncontact heart rate monitoring techniques based on millimeter-wave radar have advantages in unique medical scenarios. However, the accuracy of the existing noncontact heart rate estimation methods is still limited by interference, such as DC offsets, respiratory harmonics, and environmental noise. Additionally, these methods still require longer observation times. Most deep learning methods related to heart rate estimation still need to collect more heart rate marker data for training. To address the above problems, this paper introduces a radar signal-based heart rate estimation network named the “masked phase autoencoders with a vision transformer network” (MVN). This network is grounded on masked autoencoders (MAEs) for self-supervised pretraining and a vision transformer (ViT) for transfer learning. During the phase preprocessing stage, phase differencing and interpolation smoothing are performed on the input phase signal. In the self-supervised pretraining step, masked self-supervised training is performed on the phase signal using the MAE network. In the transfer learning stage, the encoder segment of the MAE network is integrated with the ViT network to enable transfer learning using labeled heart rate data. The innovative MVN offers a dual advantage—it not only reduces the cost associated with heart rate data acquisition but also adeptly addresses the issue of respiratory harmonic interference, which is an improvement over conventional signal processing methods. The experimental results show that the process in this paper improves the accuracy of heart rate estimation while reducing the requisite observation time.

## 1. Introduction

Heart rate is a vital sign of the presence and quality of life activity [1]; it is also one of the most important parameters for assessing vital health, and it is closely associated with stroke, sudden death, and other noncardiovascular diseases. With the accelerated aging of the global population, elderly care is facing increasing challenges. In this context, the need for new noninvasive health monitoring solutions is becoming increasingly urgent. Currently, some noncontact monitoring technologies utilize camera sensors [2,3]. These technologies can extract vital parameters, such as heart and respiration rates, via video signal processing, offering unique value in clinical and telehealth settings [4]. However, there are concerns regarding the accuracy of camera sensors in certain applications, and extended usage might pose privacy issues. Thermal sensor technology can partially address this problem [5]. However, their results can be influenced by ambient temperatures and weather conditions, limiting their applicability. Whereas Wi-Fi-based solutions can estimate vital signs more accurately [6], their design usually requires the placement of transmission and receiving antennas in different devices, which results in higher power consumption.

A radar system, a noncontact vital sign monitoring technology, is one of the frontier technologies in this field [7]. The main advantage of this technology is that it does not require any cables or electrodes to achieve noninvasive monitoring of the monitored person. By using this technology, the monitoring staff can quickly locate the monitored person in a room and perform respiratory and heart rate measurements, thus greatly improving the efficiency of the monitoring process and reducing the disturbance experienced by the monitored person. A noncontact radar heart rate monitoring system is a good solution for mitigating the inconvenience of the contact between traditional monitoring devices and the body surface while providing a more efficient and convenient alternative for detecting patients’ vital signs in traditional health care settings [8]. Among the different kinds of available radar technologies, frequency-modulated continuous-wave (FMCW)-based radar can simultaneously capture the distance and speed information of targets [9] in the radar field of view, making it suitable for the noncontact monitoring of users’ heart rates, respiration rates, and other physiological signs; additionally, it is technically easy to integrate and controllable in terms of costs, so it has critical applications in the medical health monitoring field and important application value.

As an emerging technology, millimeter-wave radar technology still faces some difficulties and challenges when monitoring human heart rates. Due to the low frequencies and small amplitudes of heartbeat signals, they are susceptible to interference from DC offsets, respiratory harmonics [10], and environmental noise [11], resulting in greater heartbeat signal extraction and identification difficulty. In addition, when millimeter-wave signals pass through human tissues, significant attenuation occurs, which decreases the strength of the received signal, further affecting the accuracy and stability of heart rate monitoring. Small movements of the human body may also lead to signal interference. For example, heartbeat and respiration signals may change due to human movement, negatively affecting the stability and accuracy of heart rate estimation.

To address the above issues, many approaches have employed machine learning and deep learning techniques to process and optimize raw radar data, rather than relying solely on traditional signal processing techniques. For example, one study employed an artificial neural network (ANN) to extract heart rate signals [12], achieving low latency and computational complexity in heartbeat detection tasks. However, this method requires a large amount of training data, including the subject’s FMCW radar signal, ECG signal, and respiratory signal, which leads to a relatively high data acquisition cost. In addition, due to the limited size of the network parameters, the adaptability of this method in different environments still needs to be further optimized. Other studies have used deep learning networks to simultaneously extract respiratory and heart rates [13], enhancing the robustness of vital sign extraction to some extent. However, if the data distribution of the target task is significantly different from that of the training data, the generalization ability of the utilized model may be challenged.

To address these problems found in existing research, this paper proposes a heart rate estimation algorithm for radar signals based on self-supervised MAE pretraining, i.e., a heart rate estimation network called the “masked phase autoencoders with a vision transformer network” (MVN) using FMCW radar. The MVN not only solves the problem of respiratory harmonic interference, which is difficult to solve with some of the existing signal processing methods [14], to further improve the accuracy of heart rate estimation but also aims to solve the problems that large amounts of labeled data are required and that laborious labeling is required for deep learning in heart rate prediction tasks. This is performed while improving the accuracy and robustness of heart rate recovery through self-supervised pretraining and migration learning methods. In addition, the method in this paper significantly reduces the required radar observation time while reducing the effects of respiratory harmonics and noise on frequency estimation and makes the heart rate estimation process more accurate with a small number of labeled samples.

Overall, the method is divided into four stages. In the subject localization stage, DC and noise reduction are performed using mean reduction to more accurately extract the positions of subjects. During the phase preprocessing stage, the phase signals corresponding to the subject positions are phase-differenced and interpolated inward to smooth out phases with excessive jump amplitudes. In the network training stage, masked self-supervised learning is performed on unlabeled radar phase data via self-supervised pretraining [15] using masked autoencoders [16]. Then, transfer learning [17] facilitates heart rate estimation on a subset of radar phase data with heart rate labels, employing the encoder segment of an MAE with the ViT network [18]. Finally, in the heart rate prediction stage, this paper uses the trained MVN in its entirety for heart rate estimation. The simulation results show that the method in this paper improves the accuracy of heart rate estimation while reducing the required observation time. Compared to signal processing methods, it has higher accuracy, and compared to previous deep learning methods, it can make heart rate estimation more accurate with a small number of labeled samples.

## 2. Related Work

### 2.1. Noncontact Heart Rate Monitoring

Traditional heart rate monitoring methods typically use electrocardiogram (ECG) technology. Nevertheless, it is challenging to satisfy the demand for prolonged monitoring due to limitations, such as the need to touch electrodes and the inconvenience of portability. In addition, some people using ECG electrodes may suffer from discomfort, such as dermatitis, which further limits the application of ECG technology. Therefore, medical institutions must adopt noncontact heart rate monitoring systems to meet the demand for long-term, efficient, and safe monitoring. Noncontact vital sign monitoring technology based on radar sensors is a research area of great interest. Compared to traditional heart rate monitoring methods, radar has many benefits. First, it is convenient for patients to use without undressing or wearing electrodes, and it avoids concerns about patient privacy. Second, the technology is suitable for both in-bed and out-of-bed situations, enabling 24 h hospital and daily monitoring. However, some aspects could be improved regarding the application of this technology. Its signal power and signal-to-noise ratio are low because electromagnetic waves suffer path losses as they pass through the body, and heart rate signals have low energy and are susceptible to interference from respiratory signals. In addition, self-motion and movement with the body can also pose challenges during radar-based vital sign monitoring. Therefore, obtaining accurate heart rate signals is a challenging problem.

To cope with these problems, the current solution strategies focus on three directions: signal processing-based methods, machine learning-based methods, and deep learning-based methods.

Among the signal processing-based approaches, Sharpe et al. [19] used FMCW radar for vital sign monitoring in 1990. In 2009, Anitori et al. [20] proposed two methods for calculating heart and respiration rates using signal amplitude and phase information. However, specific solutions for reducing harmonics and noise are inefficient. In 2013, Zhang et al. [21] extracted human heartbeat rate signals using a 24.15-GHz FMCW radar with a scanning bandwidth of 72 MHz. They proposed a projection matrix method for periodic clutter suppression. Although this approach considers the clutter interference caused by the surrounding static objects, it does not eliminate the interference of human micromotion on the heartbeat rate estimation process, so the extracted heartbeat information might be incorrect. In recent years, in the field of medical health monitoring based on FMCW radar, the research results of Dr. Fadel Adib of the Multimedia Laboratory at MIT have become widely regarded as representative work. Adib et al. [22] extracted human vital signals via the FFT operation with a Hanning window using FMCW radar-possessing frequencies of 5.46–7.25 GHz. This study demonstrated the potential of FMCW radar for use in human physiological sign monitoring as an application and spawned a series of studies on noncontact medical health monitoring techniques, but the extraction process of this method still requires a minimum signal-to-noise ratio (SNR) to distinguish vital signals from noise. Alizadeh et al. [23] obtained human vital signals through a phase data analysis using the range fast Fourier transformation (FFT), DC compensation, phase-unwrapping, and FFT vibration operations. The authors applied 77-GHz millimeter-wave radar to monitor the vital signal components by extracting the intermediate frequency (IF) signal phase. The 77-GHz millimeter-wave radar selected by the authors avoids the high harmonics caused by amplitude signal estimation. It somewhat solves the DC signal offset problem caused by quadrature receivers. Nevertheless, the method proposed in [23] may cause the phenomenon in which the respiratory harmonic frequency masks the heartbeat frequency, and it is challenging to handle signals containing more ambient noise, leading to a decrease in the accuracy of heartbeat rate estimation. Koyanaka et al. [24] performed heart rate estimation on FMCW radar using empirical mode decomposition (EMD). Before implementing EMD, the author first carried out a DC compensation procedure to minimize the errors caused by the DC component. Subsequently, EMD was utilized to isolate and extract vital signs. The method addresses the effect of the DC component to some extent but still requires a small signal-to-noise ratio to extract heartbeat signals and struggles to deal with signals containing more ambient noise.

Among the available machine learning-based methods, Malešević et al. [12] proposed a method based on continuous-wave Doppler radar and an artificial neural network (ANN) to detect individual heartbeat events. This ANN directly uses raw radar signals as input and was evaluated in an experiment with 21 healthy volunteers. The method can detect single heartbeats quickly without heavy signal preprocessing. However, the method requires a large amount of training data, which need to be acquired simultaneously with the ECG signals and respiratory signals, resulting in relatively high data acquisition costs. In addition, due to the small size of the network, its performance in different environments still needs to be further optimized. Zar et al. [25] proposed a noncontact Doppler sensor-based heartbeat detection method for this purpose, aiming at evaluating R–R interval (RRI) information without the need to attach devices. To address the peaks that may be caused by breathing and slight body movements, the authors used a peak selection method based on the Viterbi algorithm with a branching metric as the squared difference between two neighboring RRIs. Experimentally, it was demonstrated that this method produces sufficient peak detection results when combined with a spectrogram-based method. However, when dealing with signals that have a significant amount of environmental noise, there is a possibility that the accuracy may be compromised.

Among the available deep learning-based methods, Shih-Hsuan Lai et al. [26] performed noncontact exercise heart rate monitoring with a temporal convolutional network (TCN). As time progresses, DC offsets often exhibit complex change patterns. Because neural networks can effectively learn the nonlinear relationships in data, they can still learn and correct for such offsets. In addition, neural networks can learn phase data directly, thus avoiding the harmonic problems caused by respiratory and heartbeat signals that do not have exact sine–cosine waveforms. Thus, the problems of harmonics and DC offsets are better solved using neural network techniques. Such a method can achieve an average accuracy of 85% on various types of exercise equipment but is still influenced by the swing of the person’s arm. Daiki Toda et al. [27] used FMCW radar and a convolutional neural network approach to successfully reconstruct ECG signals. However, the method failed to recover the ECG waveform under certain low signal-to-noise ratio conditions.

Overall, the existing heart rate estimation algorithms are weak in terms of noise immunity, and their robustness still needs improvement. In addition, the heart rate signal estimation methods using neural networks also face the problems of high data acquisition costs and the fact that different data characteristics are possessed by different subjects.

### 2.2. Deep Learning Networks Applied in Heart Rate Estimation

Among the existing algorithms for heart rate estimation using neural networks for FMCW radar, CNNs [28] are widely used. CNNs share convolutional kernels and can handle high-dimensional data in a stress-free manner and automatically perform feature extraction. Nevertheless, single-layer convolution cannot capture long-range features, and to attain higher heart rate recognition accuracy, the depth of the CNN must be continuously increased, leading to poor CNN training efficiency and a complex and unstable training process [29]. Although RNNs [30] can handle temporal information, the RNN input of one cycle depends on the previous cycle’s output, and the parallel training ability of such a network is poor. In addition, for long sequences, memory limitations hinder the batch processing of the training samples. LSTM [31] requires four fully connected layers per unit to run in each sequence time step, and the fully connected layers require a large amount of memory bandwidth to perform computations; it is usually difficult for the system’s memory to meet this requirement. Consequently, training RNNs and LSTM poses considerable demands on computer hardware, and these methods are challenging to implement.

Combining the above reasons, this paper introduces the ideas of self-supervised pretraining and transfer learning in natural language processing [32] and proposes an MVN heart rate estimation network based on MAE self-supervised pretraining and ViT transfer learning to improve the accuracy of heart rate value estimation while alleviating the cost of collecting heart rate monitoring data.

The ViT model is attributable to the study of Dosovitskiy et al. [18]. Combining the standard transformer model [33], an attempt was made to apply it to images. The study achieved advanced performance on several image recognition tasks without using prior image knowledge, such as convolution. When trained with sufficient model parameters and training data, the ViT exhibited better performance than ResNet [29], with much better performance while spending significantly less computational resources during pretraining. Compared to the RNN model, a ViT relies mainly on matrix multiplication computations in its practical implementation, which enables parallel computation and greatly improves the training speed of the model.

The MAE model is derived from the work of Kaiming He et al. [16], which was an attempt to perform self-supervised training for computer vision based on the ViT model. The MAE model extends the training process of the ViT to unlabeled data, making it possible to pretrain and fine-tune image tasks on a large scale, similar to natural language processing, making the cost of image labeling significantly lower. By conducting masked training, the model can improve its prediction accuracy while significantly reducing the required training time. Compared to the ViT-Huge model, the MAE model achieves the same accuracy with a smaller labeled dataset.

Compared to traditional convolutional neural networks (CNNs) and recurrent neural networks (RNNs), MVN heart rate estimation networks use a self-attention mechanism to process the input data and acquire the following advantages:

Better generalization capability: The architectures of CNNs and RNNs are typically fixed. For instance, the filter size in CNNs remains constant, and RNNs sequentially process sequences of a set length. This suggests that they might not be particularly adept at handling data with varying sizes and shapes. In contrast, the self-attention mechanism can dynamically and concurrently assign weights to each element in the input data, allowing it to handle data of different sizes and structures more naturally. Moreover, networks based on the transformer architecture can often stack more layers without encountering vanishing or exploding gradients. Consequently, the MVN, which utilizes a self-attention mechanism, can more effectively manage data with diverse sizes and scales, enhancing the model’s generalization capabilities and making it more adaptable to larger or more complex tasks.

Better efficiency: Compared to CNNs and RNNs, the self-attentive mechanism can parallelize computations better when processing input data. Therefore, it performs faster in terms of its training and inference speeds.

Better cross-domain performance: The heart rate estimation network (MVN) is a transformer-based model, with its core built around a self-attention mechanism. This mechanism allows the network to process all elements in a sequence simultaneously, allocating distinct weights to each element. Such a feature enables a transformer to account for the long-range dependencies within a sequence, unlike CNNs, which are constrained by the sizes of their convolutional kernels, or RNNs, which rely on recursive sequence processing. In recent years, due to these capabilities, transformers have excelled in various domain-specific tasks, including speech recognition [34], signal processing [35], image recognition [18], and natural language processing [36]. Given that the MVN incorporates multihead self-attention and multiple stacked layers, it can harness vast amounts of data to capture intricate patterns and relationships, offering greater potential for extracting and processing radar data [13].

Combining the above advantages, the MAE network can learn a large amount of scale-free radar phase data during self-supervised learning, which results in a better ability to extract features from radar data. Subsequently, transfer learning is performed on the feature vector output by the MAE network using the ViT network to complete the mapping of the radar phase to the heart rate using less labeled data. This reduces the acquisition cost while improving the robustness and accuracy of heart rate recovery.

## 3. FMCW Radar-Based Heart Rate Estimation Principle

The original heart rate signal acquisition process derived from FMCW radar theory [23] is shown in Figure 1. The right side is the monitored human body. The left side shows the FMCW radar device, which contains the FMCW radar generator, transceiver antenna, mixer, analogue-to-digital converter, and signal processing module.

First, the FMCW radar signal generator periodically generates linear FM pulses [37] as a (Chirp) signal *C_TX_*(*t*) and transmits it through the antenna *TX*, whose frequency increases linearly with time, as shown in Equation (1):(1)$$C_{TX}\left(t\right)=A_{TX}exp\left(2\pi f_{c}t+\frac{\pi Bt^{2}}{T_{c}}\right)$$

*A_TX_* and *f_c_* are the amplitude and starting frequency of the signal, respectively; *B* and *T_c_* are the signal bandwidth and duration, respectively, and *t* is the time variable of the signal.

The antenna *RX* accepts echo signals from multiple objects to form the received signal *C_RX_(t*), as shown in Equation (2):(2)$$C_{RX}\left(t\right)=\overset{n}{\underset{i=1}{\sum }}r_{i}C_{TX}\left(t-\frac{2l_{i}}{c}\right)=A_{TX}\overset{n}{\underset{i=1}{\sum }}r_{i}\exp\left[2\pi f_{c}\left(t-\frac{2l_{i}}{c}\right)+\frac{\pi B}{T_{c}}{\left(t-\frac{2l_{i}}{c}\right)}^{2}\right]$$

*c* is the speed of light, where *l_i_* and *r_i_* are the distance from the object to the radar in the *i-*th distance cell and its reflection coefficient, respectively. After obtaining the echo signal, the mixer mixes the transmitted signal *C_TX_(t*) and the received signal *C_RX_(t*) and outputs the IF signal *Y_IF_(t*), as shown in Equation (3):(3)$$Y_{IF}\left(t\right)=A_{TX}\overset{n}{\underset{i=0}{\sum }}r_{i}\exp\left(\frac{4\pi Bl_{i}}{cT_{c}}t+\frac{4\pi f_{c}l_{i}}{c}+\frac{4\pi Bl_{i}^{2}}{c^{2}T_{c}}\right)\approx A_{TX}\overset{n}{\underset{i=0}{\sum }}r_{i}\exp\left(\overset{fi}{\overset{︷}{\frac{4\pi Bl_{i}}{cT_{c}}t}}+\overset{\varphi_{i}}{\overset{︷}{\frac{4\pi f_{c}l_{i}}{c}}}\right)$$

*A_TX_* is the amplitude of the IF signal, and *f_i_* and *φ_i_* are the frequency and phase of the reflected signal at the *i-*th distance unit, respectively. From part *f_i_* of the above equation, we can see that the objects reflected at different distances produce different frequencies in the IF signal, so the FFT transform of the IF signal can distinguish the different reflected objects. The minimum distance at which the FMCW radar can distinguish different targets, i.e., the distance resolution *l_res_*, is shown in Equation (4):(4)$$l_{res}=\frac{c}{2B}$$

After confining the reflected object to the human body, the vital signals of the human body cause the thorax to undergo small regular vibrations. In turn, the vibration of the human thorax can be described as *l_i_(t*) = *l_i_ + h_i_(t*), where *l_i_* is the distance from the human body to the radar and *h_i_(t*) is the small-amplitude regular vibration of the thorax over time. At this point, *φ_i_* in Equation (3) is equivalent to:(5)$$\varphi_{i}\left(t\right)=\frac{4\pi f_{c}}{c}\left[l_{i}+h_{i}\left(t\right)\right]=\frac{4\pi }{\lambda_{c}}\left[l_{i}+h_{i}\left(t\right)\right]$$

At this point, the phase in the IF signal becomes a function of time *t*, where *λ_c_* is the wavelength of the FMCW radar signal. Based on this function, *h_i_(t*), the regular vibration of the human chest, can be obtained. Thus, by extracting the phase information of the IF signal and processing it, the heartbeat signal corresponding to the human body can be obtained.

## 4. Flow and Design of the Heart Rate Estimation Algorithm

The structure of the millimeter-wave radar heart rate estimation algorithm based on self-supervised transfer learning proposed in this paper is shown in Figure 2.

It mainly consists of four parts: person localization, phase preprocessing, MVN training, and heart rate estimation. In the person localization stage, the person location is obtained using the FFT with the Hanning window function with mean decimation. During the phase preprocessing stage, the initial phase signal is received using the inverse tangent function. Subsequently, phase differencing, phase unwrapping, and interpolation smoothing are applied to obtain a cleaner phase signal. In the MVN training stage, masked self-supervised training is performed on the radar phase data using the MAE network. Subsequently, a small amount of data with heart rate markers are used for transfer learning training during the transfer learning process of heart rate estimation. The final heart rate estimation step is performed using the MVN to obtain a heart rate estimate for the window corresponding to the input phase data.

## 5. Subject Positioning and Phase Preprocessing

This step uses the FFT with a Hanning window [38] and mean reductions to precisely locate the person of interest’s position. After locating the person’s position, the phase signal containing the heart rate signal is extracted. The signal is phase-unwrapped, phase-differenced, interpolated, and smoothed to obtain a cleaner radar phase signal.

### 5.1. FFT and Mean Reductions for Locating Subjects 

Because the phase information of the target is contained in the IF signal, this step mainly calculates the distance unit at the location where the FFT is located in the human chest cavity. However, due to the presence of DC and static clutter in an IF signal, it is easily confused with the echo signal generated by the human thorax, thus affecting the accuracy of thorax localization. Therefore, the FFT with the Hanning window function and the mean reduction method is used here to reduce its influence and improve the accuracy of human localization as follows:

The Hanning window function is first added to each frame of the IF signal, and subsequently, the FFT is performed on each frame to obtain a long time–frequency matrix. From the *f_i_* part of Equation (3), the frequency of the object in the IF signal is proportional to its distance. Hence, the position of the peak in the frequency directly corresponds to the distance cell where the current target is located. Subsequently, the complex data in each row of the distance slow-time matrix are modulated, and the modal value of this row represents the reflected power at a fixed distance cell. The distance slow-time maps without DC, clutter removal, static objects, human thorax, and distance cell where the zero frequency is located all have high reflection values. Because the vital signs have periodicity, the distance unit data where the chest cavity is located also have periodicity. In contrast, the multipath reflections from static objects and the environment have little or irregular motion. DC signals have no phases, and the data distribution of the distance units where they are located is not periodic. Therefore, this paper proposes the averaging-based reduction method to eliminate the effects of static clutter and DC by subtracting the average value at each distance, i.e., each row in the matrix. The calculation process is shown in Equation (6):(6)$$du_{ij}^{’}=du_{ij}-\frac{1}{n}\sum_{j=1}^{n}du_{ij}$$
where *du_ij_* denotes the data in row *i* and column *j* of the matrix and *du′_ij_* represents the data in row *i* and column *j* of the matrix after executing the mean reduction method.

Figure 3 illustrates the schematic operation used to optimize the subject localization step using the mean reduction method. After performing the FFT on each frame, a distance slow-time map is obtained. In this map, static objects, human thoracic cavity, and distance units at zero frequency all exhibit higher reflection values. Subsequently, this paper applies the mean reduction method based on Equation (6) to optimize each row of the matrix obtained after implementing the range FFT. Figure 3 also provides a comparison between the signals observed before and after optimization.

After applying mean reduction and de-DC to the long time–frequency matrix, the human chest signal with its periodicity is preserved, and the effects of static clutter and DC are diminished. Subsequently, a peak frequency localization operation is performed to obtain the distance unit where the human chest cavity is located.

### 5.2. Extracting the Phase of the Distance Unit Where the Person Is Located

This step works by extracting phase information at this distance after obtaining the distance unit where the human chest cavity is located by localizing the long time–frequency matrix, restoring the human chest displacement, and further optimizing the phase using phase unwrapping, phase differencing, and interpolation smoothing operations.

For the complex data contained in the long time–frequency matrix, the phase changes continuously with time, which is expressed in the matrix as the phase changing constantly with each frame of the IF signal. In contrast, each row in the matrix corresponds to one frame of the IF signal. The phase change *φ_wrap_*(*p*) of the target over time is obtained by extracting the phase of the distance unit where the human body is located using the inverse tangent function, as shown in Equation (7):(7)$$\varphi_{wrap}\left(p\right)=\arctan\left(\frac{Q\left(p\right)}{I\left(p\right)}\right)$$
where *Q*(*p*) and *I*(*p*) are the imaginary and real parts of the *p-*th frame of the complex variable signal in the distance cell where the chest cavity is located, respectively. Due to the nature of the inverse tangent function, the derived phases are both wrapped in the interval [−*π*, +*π*]. However, the physical displacement of the human chest cavity is greater than *λ_c_*/4, so the phase change in Equation (5) exceeds the [−*π*, +*π*] interval. In this case, a correction must be made by adding or subtracting *2π* in the phase-unwrapping operation. This operation is computed as shown in Equation (8) and is performed only for phases that vary beyond the interval [−*π*, +*π*].
(8)$$\begin{array}{ccc} \varphi \left(p+1\right) & = & \{\begin{array}{c} \varphi_{wrap}\left(p+1\right)-2\pi \\ \varphi_{wrap}\left(p+1\right)+2\pi \\ \varphi_{wrap}\left(p+1\right) \end{array}\begin{array}{cc} & \mathrm{if}\;\varphi_{wrap}\left(p+1\right)-\varphi_{wrap}\left(p\right)>\pi \\ & \mathrm{if}\;\varphi_{wrap}\left(p+1\right)-\varphi_{wrap}\left(p\right)<-\pi \\ & \mathrm{otherwise} \end{array} \end{array}$$

The phase variation *φ*(*p*) exhibited by the human thorax after the unwrapping operation is thus obtained. According to the functional form of Equation (5), the complex signal produced after unwrapping can be expressed as Equation (9). In this equation, *l_chest_* is the distance from the human thorax to the radar, while *h_chest_*(*p*) describes the regular vibration of the human thorax. Specifically, *h_chest_*(*p*) encompasses the thoracic cavity vibrations induced by both breathing and heartbeats. Breathing results primarily from the expansion and contraction of the lungs, while heartbeats arise from the rhythmic pulsation of the heart. Each of these vibrations possesses distinct frequencies and amplitudes, yet they exhibit pronounced nonlinearity.
(9)$$\varphi \left(p\right)=\frac{4\pi }{\lambda_{c}}\left[l_{chest}+h_{chest}\left(p\right)\right]$$

To eliminate the baseline phase drift caused by body movements, it is necessary to perform a differential operation on the phase after the untwisting process, i.e., to determine the difference between two consecutive phase values to obtain a phase differential signal, which can be performed to eliminate the phase drift while preserving the heart rate signal. The phase difference is calculated as shown in Equation (10):(10)$$\varphi_{dif}\left(p\right)=\varphi \left(p+1\right)-\varphi \left(p\right)$$

Because thoracic micromotion makes the phase change according to a nonlinear law, the irregular noise in space can quickly destroy the continuity of the phase signal change. When the phase difference exceeds a certain threshold, it is considered that the phase here jumps due to a disturbance, and the inward interpolation method can be used to smooth the phase with an excessive jump amplitude. The phase *φ_insert_*(*p*) obtained after interpolation is shown in Equation (11):(11)$$\varphi_{insert}\left(p\right)=\frac{\varphi_{dif}\left(p-1\right)+\varphi_{dif}\left(p+1\right)}{2}$$

The above work enables the heart rate prediction network (MVN) to always receive cleaner heart rate phase signals, further enhancing the robustness of the heart rate estimation process and reducing the influences of the surrounding environment and subject anomalies on the phase while preserving the heart rate component.

## 6. Heart Rate Prediction Algorithm Based on the MVN

### 6.1. Overall MVN Architecture

To solve the problems of the existing FMCW radar method in heart rate monitoring, a heart rate estimation network (MVN) based on FMCW radar is proposed in this paper. The network is based on an MAE network with a ViT transfer learning network as a post-network. The MAE encoder has a better feature extraction ability for the phase data encountered during the learning process, and the self-supervised pretraining method enables the deep network to learn autonomously on unlabeled data. Based on this, the ViT network is used as the post-transfer learning module to improve the accuracy and robustness of heart rate estimation while reducing the cost of labeled data acquisition and preserving the feature extraction capability of the MAE network.

The training process of the network is divided into two phases. In the self-supervised pretraining stage, the network uses the MAE network for the masked self-supervised learning of unlabeled radar phase data; during the transfer learning phase, the masked self-encoder part of the MAE is used as the MVN encoder, and the ViT network is used as the MVN. In the transfer learning stage, the masked self-encoder part of the MAE is used as the MVN encoder, and the ViT network is used as the MVN decoder to perform transfer learning on a small amount of phase data with heart rate markers.

The MVN uses an asymmetric encoder–decoder architecture in both stages, where the number of encoder layers is larger than the number of decoder layers. Such an asymmetric coder–decoder architecture can reduce the time consumption of transfer learning while retaining more parameters of the self-supervised pretrained model. The architecture of one of the deep MVN subnetworks is shown in Figure 4.

The unlabeled phase data are converted into a radar phase signal matrix in the self-supervised pretraining stage. Some phase signals in this matrix are subsequently masked and labeled, identifying which signals need to be predicted or ignored. Next, the unmarked signal matrix is fed to the MAE encoder for forward propagation. After the signals pass through the encoder, marker vectors are added to the output semantic vectors in their original order and then sent to the MAE decoder to recover the actual radar phase signal matrix. This is a process of recovering from the encoded semantic space to the original signal space and is an essential step in self-supervised learning. After training and convergence are completed, the parameters of the MAE encoder are extracted and saved for the subsequent transfer learning phase.

During the transfer learning phase, a small amount of phase data with heart rate markers are used for training. The signal matrix is first fed to the MAE decoder to obtain an output semantic vector, which is subsequently provided to the ViT network. In the ViT network, the output head vector is processed by an MLP head to output the corresponding window heart rate value. In the above transfer learning process, the method in this paper locks the gradient of the MAE encoder network. That is, the method in this paper does not train the encoder but only the ViT network. This operation can keep the parameters of the encoder unchanged to utilize the feature representations learned during the self-supervised pretraining stage. It also focuses on tuning the ViT network to better adapt it to the target task.

In addition, as seen in the phase data shown in Figure 4, the person’s location does not occupy more than one column in the long time–frequency matrix. However, in traditional signal processing strategies, only the frequency with the strongest amplitude is usually selected to estimate the person’s position. This method, although simple, may need to pay attention to other helpful information. Instead, taking advantage of a transformer, this paper selects five range bins centered on the person’s position, together with the number of frames in the time window, and passes them into the heart rate prediction network (MVN) in the form of a two-dimensional matrix. This approach can fully use the phase data information to enhance the accuracy and robustness of heart rate estimation.

### 6.2. MAE-Based Self-Supervised Pretraining

The asymmetric encoder and decoder of the MAE contain multiple transformer encoder layers, where each of these layers uses layer normalization [39]. A schematic diagram of the layer normalization module is shown in Figure 5.

Layer normalization (LN) first calculates the mean and variance of each feature for each sample, unlike batch normalization (BN) [40], which normalizes each feature of each batch. It first computes the mean and variance of each feature of each batch and then normalizes each feature. Compared to BN, LN avoids performance problems involving small batches while not requiring normalization for each channel as instance normalization (IN) [41] does.

Compared to BN, LN has the advantage of avoiding performance problems on small batches while not requiring the same level of performance as instance normalization (IN). Normalization is performed for each channel. In addition, LN can be used in networks, such as RNNs and transformers, to improve their training speeds and accuracies because it does not depend on the size and order of the given batches.

The MAE network model is shown in Figure 6.

An MAE is an autoencoding network model that reconstructs the original signal from partial observations. Like all autoencoders, the method in this paper has an encoder that maps observations to semantic vectors and a decoder that reconstructs the original signal from the semantic vectors. Unlike traditional autoencoders, we adopted an asymmetric design such that the larger encoder operates only on a portion of the observed signal (without the mask tokens), while the lighter decoder reconstructs the complete signal from the semantic vectors and mask tokens. In the actual computation process, this paper used an MAE encoder containing eight transformer encoder layers and an MAE decoder containing five transformer encoder layers.

The MAE encoder includes an embedding layer, class tokens, a position embedding layer, a mask layer, a multilayer transformer encoder layer, and LN. The decoder consists of an embedding layer, a mask token layer, a position embedding layer, a transformer encoder layer, LN, and a linear layer.

The input feature dimensionality of the transformer encoder layer of the MAE encoder is 128, while the input feature dimensionality of the MAE decoder is 64. The number of multihead attention mechanisms used by the transformer encoder layers of both coders is 16, and the dimensionality of the feedforward network model is 2048. Finally, the output vector dimensionality of the linear layer is the same as that of the phase data, which is used to predict the phase signal being masked for processing.

First, the MAE encoder performs embedding coding on the radar phase data, and after adding a learnable embedding header vector and a positional embedding, it performs the masking process and inputs the data into the transformer encoder layer. In the masking layer, the authors of this paper chose a masking ratio of 40%, i.e., 40% of the unlabeled phase data are randomly marked by the masking operation, while the added learnable embedding header vectors are not masked in this operation. The rest of the data that are not masked and labeled are output as semantic vectors via LN after the multilayer transformer encoder layer computation.

Subsequently, the MAE decoder complements the input semantic vectors with corresponding learnable mask vectors (mask tokens) and positional embeddings in the order in which they are masked. Then, the recovered raw phase data are output through the multilayer transformer encoder layer, LN, and a linear fully connected layer. The training process uses the adaptive moment estimation (Adam) optimizer [42] and(MSE) statistical method to select the optimal model parameters for transfer learning after several rounds of training.

### 6.3. Heart Rate Estimation Based on ViT Transfer Learning

After completing the self-supervised pretraining process, this paper added the ViT decoder after the MAE encoder part to output the corresponding heart rate estimates, and the architecture diagram is shown in Figure 7.

In the ViT transfer learning network, the data output by the MAE encoder are first processed with embedding encoding and positional embedding, followed by LN of the multilayer transformer encoder layer. For a better transfer learning effect in heart rate prediction, the ViT transfer learning network only performs MLP head calculation on the extracted embedding header vectors, which reduces the computational effort while obtaining heart rate values estimated in the current window. Among them, the ViT network uses a five-layer transformer encoder layer, a single transformer encoder layer with an input feature dimensionality of 64, 16 multihead attention mechanisms, 2048 feedforward network model dimensions, and LN for internal normalization.

During the transfer learning training process, the model parameters of the MAE encoder are fixed and not trained. For the ViT network, the optimizer still uses the Adam method, the error statistics are MSE values, and a small amount of radar phase data with heart rate markers are used to conduct supervised training on the ViT network. After several rounds of training, the optimal model parameters are selected. The radar acquisition data are subsequently input into this network according to the processing flow, and the heart rate estimates corresponding to the current phase data are calculated.

## 7. Simulation Test

### 7.1. Experimental Design and Parameter Settings

The hardware system used for the experiments consisted of two modules: Module I was a millimeter-wave radar platform consisting of an IWR1642BOOST radar sensor evaluation board and a DCA1000EVM radar data acquisition board from Texas Instruments. Module II was an Neulog Pulse Sensor NUL-208 sensor from Neulog, which was used to collect the actual heart rates of the subjects. The equipment used is shown in Figure 8.

The experiment was conducted in a study setting with the subject sitting quietly at approximately 0.5–1.0 m in front of the radar, maintaining a natural breathing state. During the test, the heart position on the chest was kept in a substantially vertical relationship with the transmission direction of the radar antenna. During heart rate monitoring, the FMCW radar and the body-worn heart rate sensor simultaneously measured the subject. Figure 9 shows a schematic diagram of the heart rate and radar measurement experiment.

The FMCW radar parameters were set as shown in Table 1. The experiments were conducted using the measurements of the Neulog Pulse Sensor NUL-208 as reference values.

In this paper, a total of 17 volunteers were measured, including 8 females and 9 males, and each volunteer measured 20 sets of data, each set lasting for 60 s with a total of 3000 frames. The heart rate estimation window length was divided into 2.56 s and 5.12 s. Finally, 30,060 and 28,724 heart rate estimation windows were generated, which were divided into training and test sets at a ratio of 8:2. During the self-supervised pretraining stage, the entire training set was used for self-supervised training; during the transfer learning stage, a small portion of the training set was labeled with heart rate information, and the rest was unlabeled; i.e., a small portion of the training set was used for transfer learning training with labeled heart rate data. After completing the above training process, the MVN was finally evaluated on the test set.

Five comparison experiments were designed on the same test set with the same window length to test the performance of the method in this paper.

The first set of experiments verified the effectiveness of the method used in this paper in the subject location stage and compared the effects of the long time–frequency matrix before and after utilizing the mean reduction method.

The second set of experiments verified the effectiveness of the method used in this paper in the phase preprocessing stage and compared the effects of phase preprocessing before and after this operation.

The third set of experiments verified the effectiveness of masked self-supervised pretraining, and the network model with the pretraining module removed was chosen for comparison with the method in this paper. The same test sets were determined at 2.56 s and 5.12 s observation times, and the network’s convergence speeds and heart rate estimation accuracies achieved with and without the pretraining module were compared.

The fourth set of experiments was a validation experiment used to verify the effectiveness of ViT transfer learning, and the network model without ViT transfer learning was chosen for comparison with the method in this paper. The head vector was extracted at the end of the network model without ViT migration learning, and a linear layer was added for the heart rate output. The same test sets were determined at 2.56 s and 5.12 s observation times, and the convergence speeds and heart rate estimation accuracies achieved by the network with and without ViT transfer learning were compared.

The fifth set of experiments compared the heart rate estimation accuracies of multiple algorithms on test sets with different labeled data densities under 2.56 s and 5.12 s observation times. The person localization and phase extraction stages remained consistently processed, and the comparison methods only varied in the heart rate estimation stage. Data densities of 32%, 24%, and 16% were selected for different markers. The comparison algorithms are as follows:(1)The classic FMCW radar life signal estimation algorithm [22] was used, and operations, such as RangeFFT, DC compensation, and VibrationFFT, were used in this process, where the phase preprocessing step was kept consistent with that in this paper.(2)Koyanaka et al. [24] Operations, such as RangeFFT, DC compensation, and EMD modal decomposition, were used in this process, where the phase preprocessing step was kept consistent with that in this paper.(3)Toda et al. [27], i.e., heart rate estimation using CNNs on a dataset with markers. This method was used only in the training stage of the MVN, and the rest of the operations were kept consistent with those in this paper.(4)Lai et al. [26], i.e., heart rate estimation using a TCN on a dataset with markers. This method was used only in the training stage of the MVN, and the rest of the operations remained consistent with those in this paper.(5)The MVN using only labeled data. No unlabeled data were used in the self-supervised pretraining stage, and only the part of the data with heart rate labels was used in both learning stages. This method was used only in the training stage of the MVN, and the rest of the operations were consistent with those in this paper.

The error functions utilized during network training were all measured according to the MSE; given a set of true values and a set of predicted values, the MSE was calculated as shown in Equation (12):(12)$$\mathrm{MSE}=\frac{1}{m}\overset{m}{\underset{i=1}{\sum }}{\left(y_{i}-\hat{y_{i}}\right)}^{2}$$
where *m* is the number of data points and *y_i_* is the true value of the *i-*th data point. $\hat{y_{i}}$ is the predicted value of the *i-*th data point. The MSE measures the average squared difference between the predicted and true values, with smaller values indicating that better predictive performance is achieved by the model.

The performance evaluation metric for the heart rate estimation task used the average absolute error percentage (AAEP) statistical metric to measure the magnitude of the error between the measured and reference values. The AAEP is defined as shown in Equation (13):(13)$$\mathrm{AAEP}=\frac{1}{N}\overset{N}{\underset{n=1}{\sum }}\left|\frac{f\left(n\right)-F\left(n\right)}{F\left(n\right)}\right|$$
where *N* is the total number of division windows, *f*(*n*) denotes the heart rate estimate for the nth window, and *F*(*n*) denotes the heart rate reference value of the contact device for the *n-*th window.

### 7.2. Experimental Results and Analysis

(1)The first group of experiments

In this experiment, a single set of 60 s data was used for comparison purposes, the FFT obtained the long time–frequency matrix for each row, and its spectrum was plotted as shown in Figure 3. The horizontal axis is the frequency dimension, and the vertical axis is the time dimension—one of the original matrices obtained after applying the range FFT is shown in Figure 10a. The matrix was obtained after utilizing the mean decimation method for this long time–frequency matrix, as shown in Figure 10b.

From Figure 10a,b, the DC component that should have been stored near 0 Hz is effectively eliminated, the weaker background noise is eliminated, and the static object noise intensity at 15–19 Hz is further reduced. The echo signal reflected for the human body near 3–4 Hz is preserved after processing and is more prominent than the average signal observed before the reduction, so it can locate the person’s position more easily.

(2)The second group of experiments

The first part of this experiment used a single set of 60 s data for comparison purposes. Figure 11a–c correspond to the three operations in the phase and processing steps, i.e., phase extraction, phase unwrapping, and differential smoothing.

First, phase extraction was executed by computing the data from the columns corresponding to the positions of the individuals in the long time–frequency matrix. This was achieved by using the inverse tangent function as described in Equation (7), and the resulting phase signal is depicted in Figure 11a. Subsequently, a phase-unwrapping operation was performed on this signal using Equation (8) for the phase signal wrapped in the [−π, +π] interval, and then the phase signal was obtained as shown in Figure 11b. Finally, differential smoothing was performed on this signal; i.e., computing was performed using the phase differential (Equation (10)) and interpolation smoothing (Equation (11)) to obtain the phase signal, as shown in Figure 11c. The horizontal axis in Figure 11 is the time dimension, and the vertical axis is the phase angle.

The before-and-after comparison shown in Figure 11a,b demonstrates that phase unwrapping makes the phases of the parcels extracted by the inverse tangent function in the [−*π*, *+π*] interval return to the normal range, but the baseline drift caused by respiration and noise remains. The before-and-after comparison in Figure 11b,c shows that phase differencing and difference smoothing not only eliminate the baseline drift but also make the periodic variation pattern of the signal suppress the respiratory signal and enhance the heart rate signal. Overall, looking at the before-and-after changes in Figure 11a,c, the overall periodic variation exhibited by the signal is more stable, and the overall amplitude jump is reduced. Upon obtaining the phase signal after phase preprocessing, the phase signal is passed to the heart rate prediction network (MVN).

In the second part of this experiment, 32% of the phase data with heart rate markers were selected from the whole dataset for training, and the error was verified using the same test set. Table 2 gives the error comparison data yielded by the algorithm and the method proposed in this paper with and without phase differencing and interpolation smoothing.

As seen in Table 2, the accuracy of heart rate estimation is improved after performing the phase preprocessing stage with the phase differencing and differential smoothing operations. Phase differencing eliminates the linear trend in the phase signal, while differential smoothing is performed to reduce high-frequency noise. The combination of these two operations enables the method in this paper to obtain a purer radar signal, which provides better data features for the subsequent heart rate estimation procedure executed by deep neural networks.

In addition, the MAE encoder, which has phase differencing and interpolation smoothing processes, has a minor mean squared error (MSE) in terms of recovering phase data compared to that induced on the original data. This indicates that the phase preprocessing operation makes the deep neural network receive a purer radar signal, which enhances the ability of the MAE encoder to precisely extract the radar phase signal, thus further enhancing the accuracy and robustness of heart rate estimation.

(3)The third group of experiments

This experiment used 32% of the phase data with heart rate markers from the whole dataset for training, and the same test set was used to verify the error. Comparisons were made only between model versions with and without the MAE decoder. The learning rate was set to 5 × 10^−5^, and the training process was stopped after 2000 epochs. The MSE convergence curve of the MVN with the self-supervised pretrained MAE decoder removed is shown as the orange line in Figure 12. The MSE convergence curve of the MVN network with the self-supervised pretrained MAE decoder is shown as the blue line in Figure 12. Figure 12a represents the MSE under the 2.56 s window, and Figure 12b represents the MSE under the 5.12 s window. Each MSE value here denotes the mean squared error between the heart rate predicted by the network and the actual heart rate captured through the contact device on a given dataset. During training, an epoch refers to a complete traversal of the entire training dataset. The horizontal axis in the figure represents the number of epochs, and the vertical axis represents the MSE loss.

Figure 12a shows that whether the MAE decoder part is removed has little effect on the MSE convergence speed before 800 epochs. However, after 800 epochs, the network error induced with the self-supervised pretrained MAE decoder can still converge further, and this version performs better than the unsupervised pretrained network. A similar performance comparison is produced in Figure 12b with 400 epochs as the cut-off, indicating that the MAE decoder performs better in terms of generalizing the heart rate estimation process.

Table 3 gives the mean absolute error percentages of the heart rates obtained with the MAE decoder-free algorithm and the method in this paper.

As shown in Table 3, the MVN with the MAE decoder has a minor heart rate estimation error.

(4)The fourth group of experiments

This experiment used 32% of the phase data with heart rate markers from the whole dataset for training, and the same test set was used to verify the error and to make a comparison only between model versions with and without ViT transfer learning. The learning rate was set to 5 × 10^−5^, and the training process was stopped after 2000 training epochs. The MSE convergence curve of the MVN with ViT transfer learning removed is shown as the orange line in Figure 12. The MSE convergence curve of the MVN with ViT transfer learning is the blue line in Figure 12. Figure 13a represents the MSE under the 2.56 s window, and Figure 13b represents the MSE under the 5.12 s window. The MSE value here denotes the mean squared error between the heart rate predicted by the network and the actual heart rate captured through the contact device on a given dataset. During training, an epoch refers to a complete traversal of the entire training dataset. The horizontal axis in the figure is the number of epochs, and the vertical axis is the MSE loss.

As shown in Figure 13a,b, after 800 epochs of training, the network model with ViT transfer learning can further optimize the heart rate estimation error, and its performance is significantly better than that of the model without ViT transfer learning. The data in these two figures demonstrate that ViT transfer learning can improve the generalization ability of the heart rate estimation procedure in the MVN.

Table 4 gives the average absolute error percentages of the heart rates obtained without using the ViT transfer learning network and with the method in this paper.

As shown in Table 4, the MVN using ViT transfer learning induces lower heart rate estimation errors.

(5)The fifth group of experiments

Table 5 and Table 6 compare the classic algorithm [22], the Koyanaka et al. [24] method, the Toda et al. [27] method, the Lai et al. [26] method, and the MVN using only marker data with the approach proposed in this paper. To verify the advantages of the MVN in reducing the data acquisition cost, this study conducted multiple replications of the comparison experiments with different labeled sample densities (32%, 24%, and 16%). The mean absolute error percentages of the heart rates obtained by several schemes with the same test set are given in Table 5. The average absolute error percentages of the method in this paper are 8.10%, 7.92%, and 7.88% under different marked sample densities with 2.56 s windows. The 6.95%, 7.21%, and 7.33% values achieved under the 5.12 s samples are more accurate and better than those of the other schemes.

In addition, it can also be seen from Table 5 and Table 6 that the MVN proposed in this paper yields better heart rate estimation results with and without using only the labeled data, which indicates that the MVN has better performance in terms of discovering the pattern of heart rate signals. The method in this paper uses a large amount of unlabeled data for self-supervised pretraining to further improve its accuracy.

## 8. Discussion and Analysis

This study introduces a heart rate estimation algorithm for radar signals based on self-supervised MAE pretraining, utilizing a 77-GHz FMCW radar. The experiments were conducted in an environment with subjects seated approximately 0.5–1.0 m from the radar, with minimal movement. While this method may not match the precision of contact-based heart rate monitors, it demonstrates the feasibility of noncontact heart rate monitoring using radar. It also addresses some challenges associated with noncontact vital sign estimation using FMCW radar, such as interference from DC and harmonics and the high cost of data collection. Experimental validations show that our method achieves an average absolute error percentage of 8.10% with a sample density of 32% in a 2.56 s time window. This is a significant improvement over the results of other studies, such as the 18.57% value of Koyanaka et al. [24] and the 10.35% value of Toda et al. [27]. This underscores our method’s enhanced precision in noncontact heart rate monitoring scenarios and its capability to handle DC and harmonic interferences from FMCW radar. Moreover, within a 5.12 s time window, our approach achieves a minimal error of 6.95% on 32% of the labeled samples, further highlighting its robustness and accuracy. These precision metrics suggest advancements in addressing the high-cost data collection challenge, as we can achieve more accurate heart rate estimates in shorter times with fewer labeled samples.

Regarding the construction of the deep network in the proposed method, we drew inspiration from effective natural language processing network models, adopting an innovative heart rate estimation approach. We began by using easily accessible, low-cost radar signals derived from seated individuals as a dataset for self-supervised pretraining. This strategy endowed the model with enhanced feature extraction capabilities, setting the stage for a subsequent transfer learning step with rich semantic vectors. We then employed a limited set of high-cost data with heart rate labels for transfer learning. This step ensured the model’s ability to accurately estimate heart rates even with limited labeled samples. By integrating self-supervised pretraining with transfer learning, we successfully improved the accuracy of heart rate estimation while reducing the data collection costs and complexity of the network. Even with limited labeled samples, this method guarantees accurate heart rate estimates. Additionally, by combining certain signal processing techniques with deeper network models, we achieved good reductions in the interference caused by respiratory harmonics and noise, shortened the required radar observation time, and enhanced the overall efficiency and practicality of the network.

While this solution offers several innovative advantages, it has some limitations. The experimental environment is restricted to subjects in a seated position. Given the high distance resolution of the FMCW radar system, it is sensitive to human movement. Minor movements can easily mask heart rate signals [43]; hence, data are collected only from stationary users. In contrast, contact-based heart rate monitors can measure the heart rates of individuals in motion. Furthermore, the data used in this experiment primarily come from healthy subjects, and their generalizability to patients remains uncertain. To overcome these limitations, future work will focus on enhancing the algorithm’s robustness to slight movements, and we will consider collecting data from subjects with varying health conditions to improve the model’s adaptability and accuracy for specific patient groups.

## 9. Conclusions

This paper proposes a noncontact heart rate estimation method based on FMCW radar. In the subject localization stage, this paper uses the FFT with a Hanning window and mean decimation to increase the accuracy of chest position extraction; during the phase preprocessing stage, phase unwrapping, phase differencing, and interpolation smoothing are performed on the phase signal. These operations eliminate the baseline drift and amplitude jumps and ensure the purity and stability of the obtained heart rate signal. In the training stage of the MVN, masked self-supervised phase signal training is performed using the MAE network. Heart rate marker data transfer learning is then performed using the encoder part of the MAE network with a ViT network. The MVN model combining the MAE encoder and the ViT network can take full advantage of the feature performance achieving in the self-supervised pretraining stage without altering the encoder parameters. In this way, the ViT network is trained to better adapt to specific tasks. This approach enables the MVN model to reduce its training time consumption while enhancing the accuracy and robustness of the final heart rate estimation results. The network also solves the respiratory harmonic interference problem to a large extent. From the experimental results, the method in this paper exhibits high accuracy even within a short observation window. Moreover, it maintains excellent performance even in cases with sparsely labeled data. The method developed in this paper improves the accuracy of heart rate estimation while reducing the cost of radar data labeling, and it demonstrates good robustness in complex environments. These features make it more suitable for practical applications and provide a new noncontact vital sign monitoring solution.

To address the future focus of this work, this paper will further compare the proposed method with other contactless solutions and combine the developed algorithmic solution with different environments to adapt to different detected person states, such as people located outdoors and people exercising. Another interesting future work topic is the reconstruction of the ECG and PPG of a detected person, as many studies have attempted to determine the presence or absence of cardiovascular diseases based on ECG signals. Therefore, the potential ability of this algorithm to detect whether a user has cardiovascular disease is worth investigating. In addition, due to the episodic nature of cardiovascular disease, wearing an ECG device for a limited period of time may not be sufficient for monitoring abnormal activity in the cardiovascular system, whereas the FMCW radar has the ability to continuously monitor a patient, and cardiovascular disease detection techniques based on this device are also of great value.

## Figures and Tables

**Figure 1 sensors-23-07869-f001:**
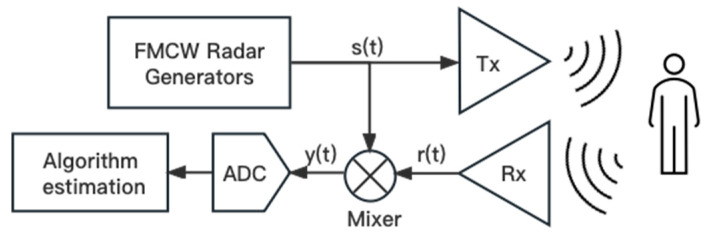
Heart rate signal acquisition principle of FMCW radar.

**Figure 2 sensors-23-07869-f002:**
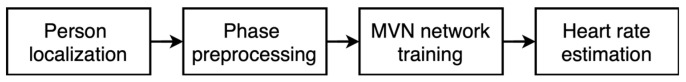
Flow chart of the heart rate estimation algorithm.

**Figure 3 sensors-23-07869-f003:**
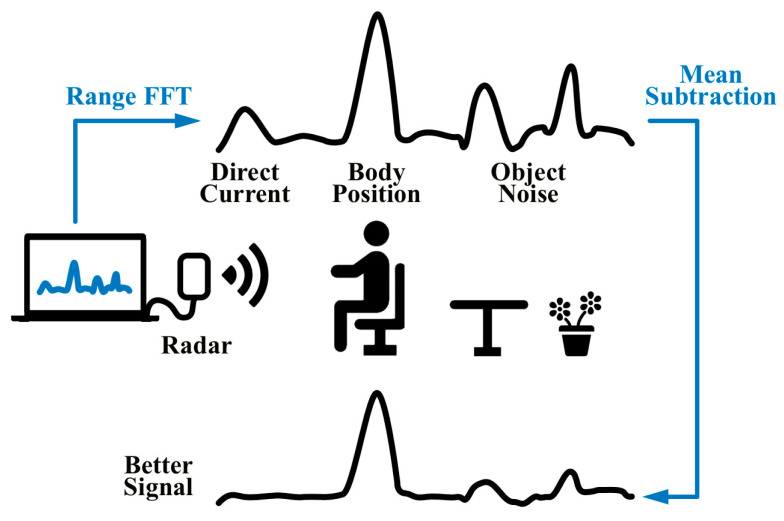
Diagram of the mean reduction operation.

**Figure 4 sensors-23-07869-f004:**
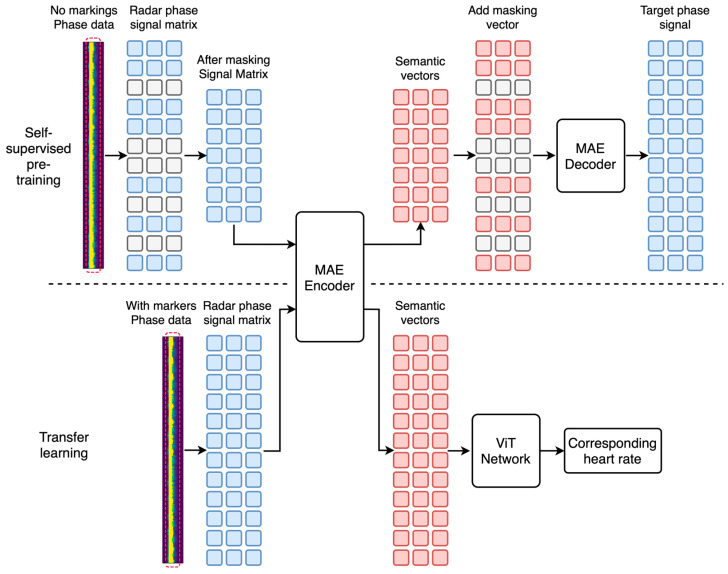
MVN architecture diagram.

**Figure 5 sensors-23-07869-f005:**
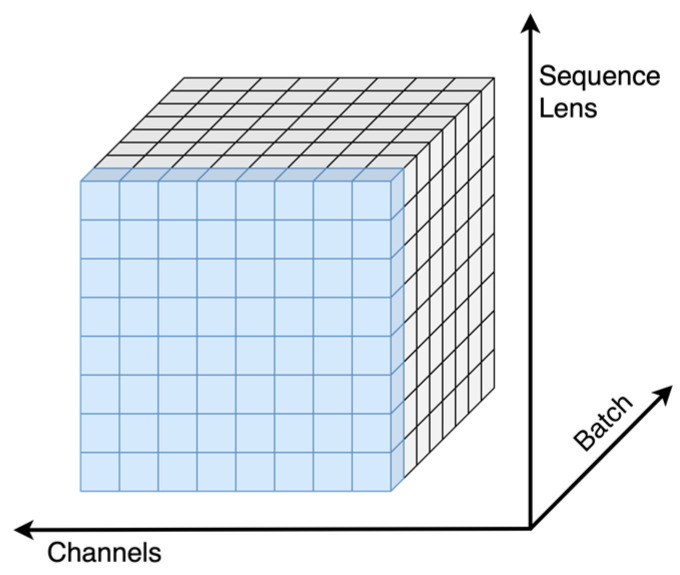
Normalization module.

**Figure 6 sensors-23-07869-f006:**
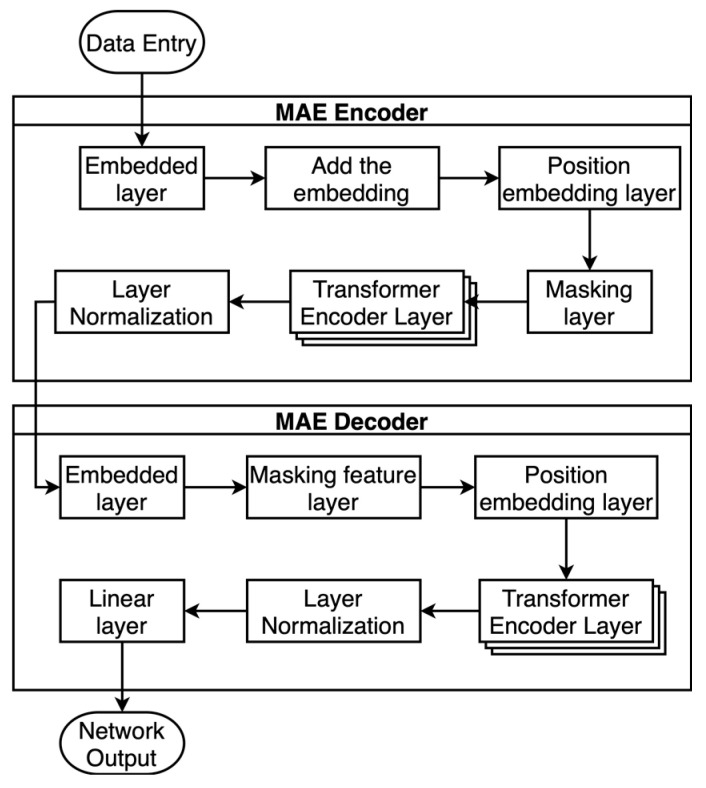
MAE network model.

**Figure 7 sensors-23-07869-f007:**
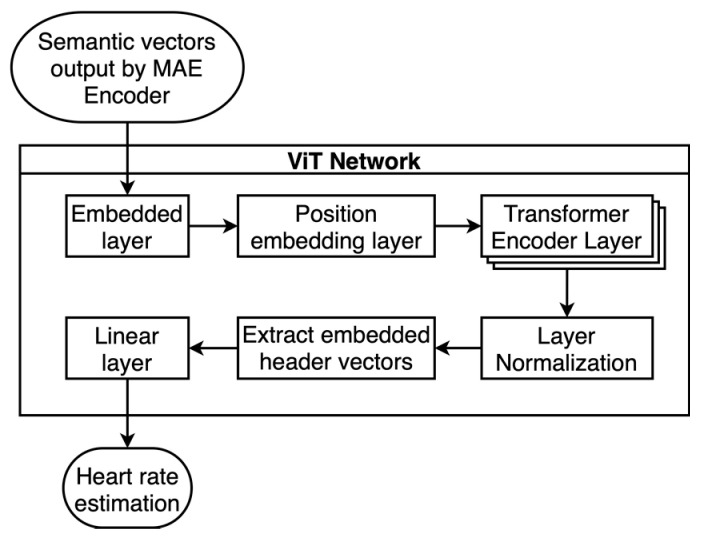
Transfer learning network model.

**Figure 8 sensors-23-07869-f008:**
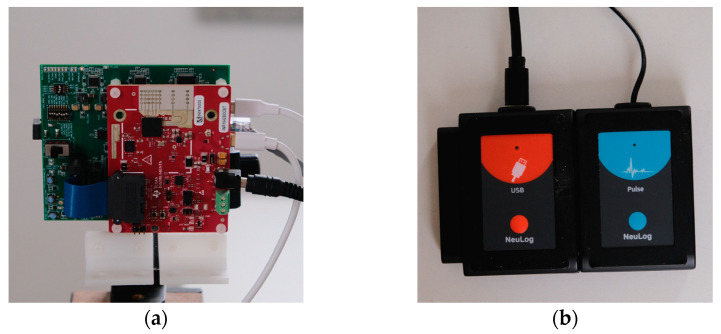
Hardware used in the experiments: (**a**) TI’s IWR1642BOOST radar sensor evaluation board and DCA1000EVM radar data acquisition board; (**b**) Neulog Pulse Sensor NUL-208.

**Figure 9 sensors-23-07869-f009:**
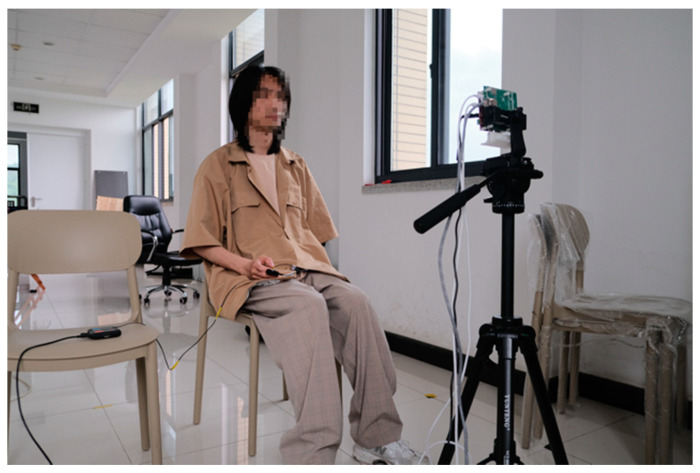
Heart rate measurement environment.

**Figure 10 sensors-23-07869-f010:**
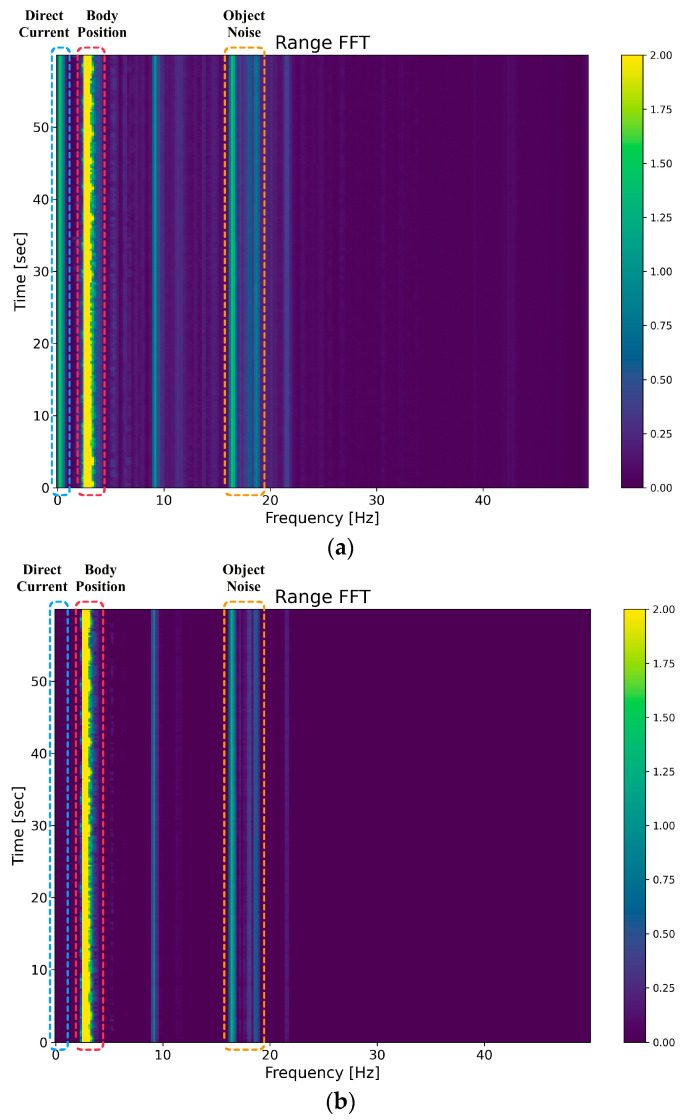
Comparison between the results obtained before and after mean normalization. (**a**) Tangent function extraction process used to obtain the signal. (**b**) Signal obtained after utilizing the mean subtraction method.

**Figure 11 sensors-23-07869-f011:**
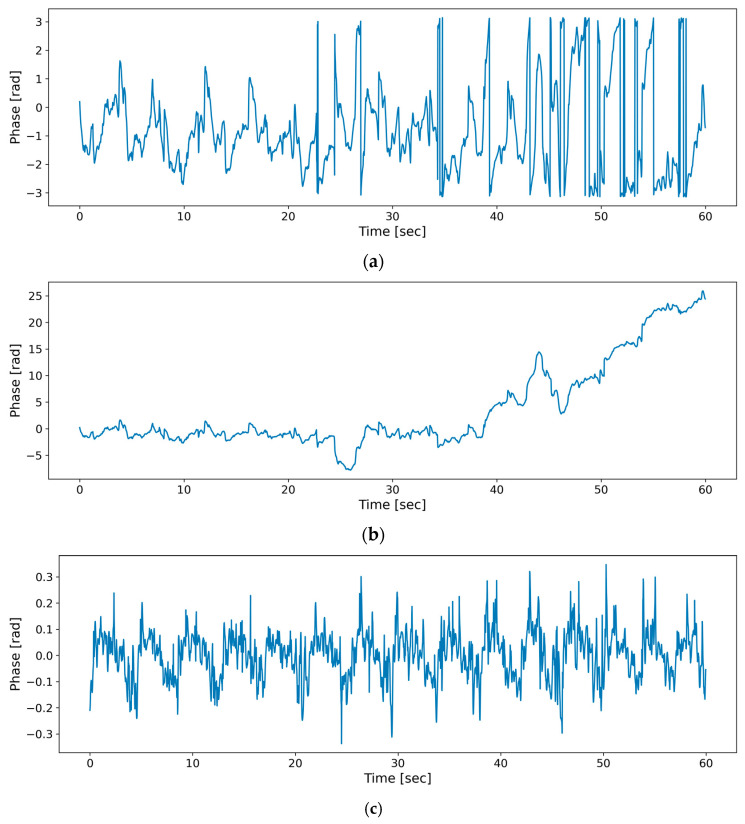
Phase signal comparison chart. (**a**) Tangent function extraction process used to obtain the phase. (**b**) Phase preprocessing stage used to obtain the signal via phase unwrapping. (**c**) Phase preprocessing stage used to obtain the signal via differencing and smoothing.

**Figure 12 sensors-23-07869-f012:**
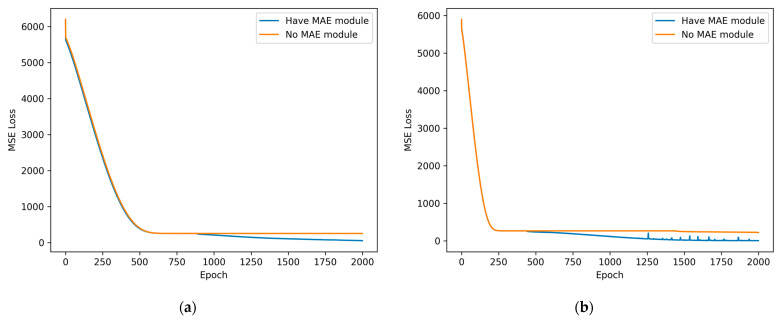
Comparison between the heart rate error convergence speeds obtained with and without the MAE decoder: (**a**) 2.56 s window MSE convergence; (**b**) 5.12 s window MSE convergence.

**Figure 13 sensors-23-07869-f013:**
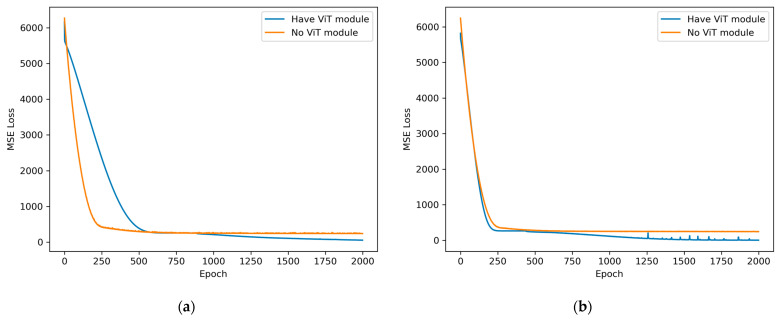
Heart rate error convergence patterns produced with and without ViT transfer learning: (**a**) 2.56 s window MSE convergence; (**b**) 5.12 s window MSE convergence.

**Table 1 sensors-23-07869-t001:** FMCW radar parameter settings.

Parameters	Value
Frequency sweep range	77–81 GHz
Use of bandwidth	3.99 GHz
ADC sampling points	256
Frame sampling rate	50 Hz
Frame rate	3000
Chirp loops	6
Period of the frame	20 ms
Number of transmission antennas	2
Number of receiving antennas	4

**Table 2 sensors-23-07869-t002:** Prediction errors of the network with and without phase differencing and interpolation smoothing.

Sampling Time	Comparison Algorithm	MAE Encoder Output MSE	Heart Rate Estimation Error (AAEP) (%)
2.56 s	No phase differentiation, interpolation smoothing	0.09231	9.91
	Methodology of this article	0.00220	8.10
5.12 s	No phase differentiation, interpolation smoothing	0.03855	9.79
	Methodology of this article	0.00215	6.95

**Table 3 sensors-23-07869-t003:** Mean absolute error percentages of the heart rates obtained with and without the MAE encoder (AAEP) (%).

Sample Time	Comparison Algorithm	32% Sample with Markers
2.56 s	No MAE decoder	10.15
	Methodology of this article	8.10
5.12 s	No MAE decoder	10.56
	Methodology of this article	6.95

**Table 4 sensors-23-07869-t004:** Mean absolute error percentages (AAEPs) of the heart rates obtained with and without ViT network transfer learning (%).

Sample Time	Comparison Algorithm	32% Sample with Markers
2.56 s	Not using ViT transfer learning	11.60
	Methodology of this article	8.10
5.12 s	Not using ViT transfer learning	10.97
	Methodology of this article	6.95

**Table 5 sensors-23-07869-t005:** Mean absolute error percentages (AAEPs) of the heart rates obtained under a 2.56 s time window (%).

Algorithm	32% Sample with Markers	24% Sample with Markers	16% Sample with Markers
Classic algorithm [22]	19.33	19.33	19.33
Koyanaka et al. [24]	18.57	18.57	18.57
Toda et al. [27]	10.35	11.40	11.68
Lai et al. [26]	9.98	8.62	9.19
MVN using only marker data	8.35	8.64	8.32
Methodology of this article	8.10	7.92	7.88

**Table 6 sensors-23-07869-t006:** Mean absolute error percentages (AAEPs) of the heart rates obtained under a 5.12 s time window (%).

Algorithm	32% Sample with Markers	24% Sample with Markers	16% Sample with Markers
Classical algorithm [22]	15.62	15.62	15.62
Koyanaka et al. [24]	17.61	17.61	17.61
Toda et al. [27]	9.67	10.28	10.47
Lai et al. [26]	8.24	8.31	8.88
MVN using only marker data	7.37	7.85	7.58
Methodology of this article	6.95	7.21	7.33

## Data Availability

The data are not publicly available due to privacy issues.

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
