# Peer review of "MAE-Based Self-Supervised Pretraining Algorithm for Heart Rate Estimation of Radar Signals"

_sensors, 2023, doi:10.3390/s23187869_

Round 1
Reviewer 1 Report
The article presents MAE-based self-supervised pretraining algorithm for heart rate estimation of radar signals of FMCW Radar. This paper introduces a radar signal heart rate estimation network named masked phase autoencoders with a vision transformer network (MVN).
The authors of the article described the methodology and presented experimental results, on the basis of which it was concluded that the process in this paper improves the accuracy of heart rate estimation while reducing the requisite observation time.
However, there are the following comments on the article.
1. Increasing the accuracy of heart rate estimation while reducing the required observation time is shown for various applications of algorithms of deep machine learning methods. However, as shown in Fig. 11 (c), the signal received at the preprocessing stage of the phase contains the heart rate explicitly and does not require further machine learning. I would like to get a comparison of the accuracy of the heart rate estimation using the machine learning algorithm proposed by the authors of the article and without it.
2. The transition from Fig. 11(a) to Fig. 11(b) is poorly described. Vertically on the axis of Fig. 11 (a-с), the units of measurement of the amplitude are not indicated. If Fig.11 (a) corresponds to equation (6), then which equations correspond to Fig.11 b, c? It is unclear what differentiation is used to obtain Figure 11? In general, Figures 11 (a-c) are not sufficiently explained, and the connection with the theoretical part of the article is not shown.
3. The use of FMCW radar to estimate the heart rate is shown only for a stationary patient, while other methods allow measuring the heart rate in motion.
Author Response
Dear Editors and Reviewers:
Thank you for your letter and the reviewers’ comments on our manuscript entitled “MAE-based self-supervised pretraining algorithm for heart rate” (ID: sensors-2535326). These comments are valuable for improving our manuscript, as well as the important guiding significance to our research. We have answered the comments carefully and made corresponding corrections. We hope the revisions meet with approval. Revised portions are marked in red on our manuscript. The main corrections in the paper and the responses to the reviewers’ comments are as follows with reviewers’ comments in red.
Q1: Increasing the accuracy of heart rate estimation while reducing the required observation time is shown for various applications of algorithms of deep machine learning methods. However, as shown in Fig. 11 (c), the signal received at the preprocessing stage of the phase contains the heart rate explicitly and does not require further machine learning. I would like to get a comparison of the accuracy of the heart rate estimation using the machine learning algorithm proposed by the authors of the article and without it.
A1: We appreciate your insights on the signal displayed in Fig. 11 (c). As you rightly pointed out, the signal received at the preprocessing stage of the phase contains the heart rate explicitly. Nevertheless, our primary objective is to harness the capabilities of deep machine learning methods to further enhance the estimation accuracy of the heart rate, especially in situations prone to interference from respiration, harmonics, and other environmental noises. In the original manuscript, we had contrasted our method with a non-machine learning algorithm, specifically:
[22] Adib, F.; Mao, H.; Kabelac, Z.; Katabi, D.; Miller, R.C. Smart Homes That Monitor Breathing and Heart Rate. In Proceedings of the Proceedings of the 33rd Annual ACM Conference on Human Factors in Computing Systems; ACM: Seoul Republic of Korea, April 18 2015; pp. 837–846.
For a more comprehensive comparison, we undertook additional experiments utilizing an algorithm from:
[24] Koyanaka, R.; Hu, Y.; Toda, T. A Study of Heart Rate Estimation on Empirical Mode Decomposition with Mm-Wave FMCW Ra-dar. IEICE Proceedings Series 2020, 63.
We have assessed and compared the heart rate estimation directly derived from the preprocessing signal against the accuracy achieved with our proposed deep learning algorithm. In Table 5, over a 2.56s window, algorithms [22] and [24] respectively exhibited absolute mean error percentages of 19.33 and 18.57 on the respective test sets. Meanwhile, in Table 6, for a 5.12s window, these values were 15.62 and 17.61, respectively. In both timeframes, the method proposed in this paper outperformed the aforementioned algorithms.
Q2: The transition from Fig. 11(a) to Fig. 11(b) is poorly described. Vertically on the axis of Fig. 11 (a-с), the units of measurement of the amplitude are not indicated. If Fig.11 (a) corresponds to equation (6), then which equations correspond to Fig.11 b, c? It is unclear what differentiation is used to obtain Figure 11? In general, Figures 11 (a-c) are not sufficiently explained, and the connection with the theoretical part of the article is not shown.
A2: We are grateful for your feedback concerning our phase preprocessing images. The paper has been amended as follows.
(1) Regarding the transition from Fig. 11(a) to Fig. 11(b): Lines 658-661 now clearly depict the link between the two figures. We employed phase unwrapping, as defined by Equation (8), on Fig. 11(a) to produce Fig. 11(b), expanding the phase wrapped within the [-π, +π] interval.
(2) Pertaining to the vertical units in Fig. 11 (a-c): We have updated and appended the unit notation on the vertical axis, denoted as Phase[rad], corresponding to phase angles.
(3) Concerning the relation of Fig.11 with equations: Beyond the relationship between Fig. 11(a) and Equation (7), we have now explicitly associated Fig. 11(b) with Equation (8) derived from phase unwrapping, and Fig. 11(c) with Equations (9) and (10) acquired through differential smoothing, elaborating on this between lines 659-665.
(4) Regarding the linkage between Fig. 11 (a-c) and the theoretical section: We have revisited and reinforced the connection between the figures and the theoretical aspects, expanding on the progressive relationship among the figures and elucidating their corresponding operations and mathematical formulas between lines 654-665.
Q3: The use of FMCW radar to estimate the heart rate is shown only for a stationary patient, while other methods allow measuring the heart rate in motion.
A3: As you rightly highlighted, the FMCW radar is employed for estimating the heart rate of stationary patients in our study. Due to the high spatial resolution of FMCW radars, they are sensitive to human movements. Slight movements can easily overshadow the heart rate signal, hence our study focused on collecting data solely from stationary individuals. Nonetheless, contact-based heart rate monitoring methods might demonstrate higher adaptability in this regard. There are ongoing efforts exploring the use of FMCW radar for heart rate estimation during patient mobility, for instance:
[13] Xie, Z.; Wang, H.; Han, S.; Schoenfeld, E.; Ye, F. DeepVS: A Deep Learning Approach for RF-Based Vital Signs Sensing. In Proceed-ings of the Proceedings of the 13th ACM International Conference on Bioinformatics, Computational Biology and Health Infor-matics; ACM: Northbrook Illinois, August 7 2022; pp. 1–5.
[31] Chen, Z.; Zheng, T.; Cai, C.; Luo, J. MoVi-Fi: Motion-Robust Vital Signs Waveform Recovery via Deep Interpreted RF Sensing. In Proceedings of the Proceedings of the 27th Annual International Conference on Mobile Computing and Networking; ACM: New Orleans Louisiana, October 25 2021; pp. 392–405.
Method [13] can cope, to some extent, with minor patient movements, enhancing the precise extraction of vital signs. Yet, when the real-world scenario diverges significantly from the training data, the method might face generalization issues. On the other hand, method [31] introduces a robust vital sign monitoring system against bodily movements. Despite its ability to recover vital sign waveforms even under intensive body movements, its accuracy remains relatively suboptimal.
To render our manuscript more comprehensive, we've elaborated on these limitations and discussed potential future possibilities and research directions of employing FMCW radar under dynamic human conditions in Chapter 8, titled "Discussion and Analysis".

Reviewer 2 Report
Comments to Authors (General)
· The authors shall have to cite more relevant works in the world for giving the coincide information.
· The authors shall have to make the concrete information in “Introduction” section.
· The paper writing style shall have to be modified.
Comments to Authors (Specific)
· The authors shall have to discuss about the Viterbi Algorithm Based on Distribution of Difference of Two-Adjacent RR Intervals for Non-contact Heartbeat Detection.
· The authors shall have to modify the Flow chart of the heart rate estimation algorithm in Figure.2 with complete format and complete that flowchart for good presentation.
· The authors shall have to mention the mathematical model for Mean reduction operation diagram in Figure.3.
· The authors shall have to give more information regarding the vibration signal of the human thorax (hchest (p)).
· The detailed process shall have to be presented in the MAE network model in Figure.6.
· The authors shall have to give the specific parameters for the analyses in Figure.6 and the Transfer learning network model in Figure.7.
· The authors shall have to mention the utilized parameters for MSE Loss vs Epoch for Figure.12.
· The authors shall have to express the idea for more accuracy in that analyses.
· The limitations and recommendation of the proposed work shall have to be discuss in the separate section before conclusion.
The writing style shall have to be modified with Extensive editing of English language required.
Author Response
Dear Editors and Reviewers:
Thank you for your letter and the reviewers’ comments on our manuscript entitled “MAE-based self-supervised pretraining algorithm for heart rate” (ID: sensors-2535326). These comments are valuable for improving our manuscript, as well as the important guiding significance to our research. We have answered the comments carefully and made corresponding corrections. We hope the revisions meet with approval. Revised portions are marked in red on our manuscript. The main corrections in the paper and the responses to the reviewers’ comments are as follows with reviewers’ comments in red.
Comments to Authors (General)
Q1: The authors shall have to cite more relevant works in the world for giving the coincide information.
A1: Thank you for your feedback and suggestion. In response, we have incorporated additional references in the revised manuscript as detailed below.
(1) In Chapter 1, paragraphs 1-2:
We have added citations related to various non-contact vital sign monitoring algorithms such as camera sensors ([4]), thermal sensors ([5]), and WiFi solutions ([6]). Furthermore, we included references that discuss the applications of radar for capturing distance and velocity information of targets within its field of view ([9]).
(2) In Chapter 1, paragraph 4:
We introduced references that pertain to the use of deep learning networks for simultaneous extraction of respiratory and heart rates ([13]).
(3) In Section 2.1, paragraphs 3-4:
We have supplemented representative works in recent years on non-contact health monitoring using FMCW radar ([22]), literature related to the Empirical Mode Decomposition (EMD) algorithm ([24]), and peak selection methods based on the Viterbi algorithm ([25]).
(4) In Section 2.2, paragraph 8:
We have incorporated references that delve into the application of the Transformer algorithm across various domains, including speech recognition ([34]), signal processing ([35]), and natural language processing ([36]).
(5) In Section 7.1, lines 606-608:
We have additionally introduced a non-machine learning experimental algorithm using Empirical Mode Decomposition (EMD) algorithm ([24]). Nonetheless, the accuracy of heart rate estimation provided by our proposed deep learning algorithm still surpasses that of the aforementioned EMD method.
(6) In Chapter 8, paragraph 3:
We have cited literature that addresses how slight human movements can easily obscure heart rate signals ([43]).
We hope these additions will enrich the manuscript's content and provide a more holistic perspective.
Q2: The authors shall have to make the concrete information in “Introduction” section.
A2: We acknowledge the necessity of furnishing the "Introduction" section with more specific details. Consequently, in the revised manuscript, we have diligently articulated the background work and research processes.
(1) Pertaining to the background: In Chapter 1, Paragraph 1, we have elaborated on specific non-contact heart rate estimation techniques, such as those based on camera sensors, thermal sensors, and WiFi solutions.
(2) Concerning AI methodologies: In Chapter 1, Paragraph 4, we've added detailed information regarding the utilization of machine learning and deep learning techniques in processing radar data. For instance, certain studies have employed Artificial Neural Networks (ANN) to extract heart rate signals. However, this approach necessitates a significant amount of training data, leading to high costs, and the adaptability of the model could benefit from enhancements. Conversely, some studies have sought to use deep learning to simultaneously extract both respiratory and heart rates, which augments robustness, yet the model's generalization capabilities continue to pose challenges.
(3) About the method presented in this paper: In Chapter 1, Paragraphs 5-6, we've elaborated on the specific stages and advantages related to our proposed method. For instance, MVN can enhance the accuracy of heart rate estimation while concurrently diminishing the amount of labeled data needed for deep learning. This method predominantly consists of four stages: personnel localization, phase preprocessing, network training, and heart rate prediction.
We hope that the aforementioned augmentations and amendments offer a lucid and comprehensive backdrop, elucidating the research's significance and its rationale for the readers.
Q3: The paper writing style shall have to be modified.
A3: Thank you for your valuable feedback on our manuscript. We have thoroughly reviewed the entire paper, revising and refining ambiguous or unclear sentences to ensure a more coherent and comprehensible narrative flow. Additionally, in the Introduction, Related Work, and Conclusion sections, we have restructured portions of the narrative to provide a more logical and cohesive progression. For Figures 2, 6, 7, and 11, as well as Equations 6, 8, and 12, we have provided further clarification regarding their relevance within the manuscript, ensuring their integration is tightly coupled with the textual content. Additionally, the manuscript has been meticulously reviewed and refined by a professional English editing service to ensure clarity and coherence. We trust that these modifications and elaborations will offer readers a more lucid understanding of the presented content.
Comments to Authors (Specific)
Q1: The authors shall have to discuss about the Viterbi Algorithm Based on Distribution of Difference of Two-Adjacent RR Intervals for Non-contact Heartbeat Detection.
A1: Thank you for the valuable insight into our research. You have recommended that we explore the application of the Viterbi Algorithm, grounded in the distribution of differences between two consecutive RR intervals, for non-contact heartbeat detection. After thorough examination of pertinent literature, we concur that this algorithm indeed holds significant relevance in the realm of non-contact cardiac monitoring. To enhance our manuscript, we have incorporated a comprehensive discussion on this algorithm in Section 2.1, paragraph 4, elucidating its principles, merits, and its relation and distinction from the method we have proposed.
Q2: The authors shall have to modify the Flow chart of the heart rate estimation algorithm in Figure.2 with complete format and complete that flowchart for good presentation.
A2: We appreciate your constructive feedback concerning Figure.2. We have meticulously redesigned and refined the flow chart in Figure.2, ensuring its format is comprehensive and its content lucidly communicated, aiming to provide an improved visual representation for our readers.
Q3: The authors shall have to mention the mathematical model for Mean reduction operation diagram in Figure.3.
A3: We appreciate your valuable suggestion. For clarity, we have elaborated and explained the connection between the Mean reduction operation in Figure 3 and Equation (6) in Section 5.1, Paragraph 3 of the revised manuscript. We have reformulated the original mean() in Equation (6) to present it as an average formula based on the summation symbol. Figure 3 demonstrates the application of the mean reduction method to optimize personnel positioning and showcases the signal comparison before and after applying the mean reduction operation as per Equation (6).
Q4: The authors shall have to give more information regarding the vibration signal of the human thorax (hchest (p)).
A4: We have provided an extended explanation about the vibration hchest(p) of the thorax in lines 393-399. The hchest(p) characterizes the periodic vibrations of the human chest cavity. Within hchest(p), it encompasses vibrations caused by both respiration and heartbeats. Respiratory vibrations mainly arise due to the expansion and contraction of the lungs, while cardiac vibrations are induced by heart pulsations. Both types of vibrations possess distinct frequencies and amplitudes and are highly nonlinear in nature.
Q5: The detailed process shall have to be presented in the MAE network model in Figure.6.
A5: To offer a more comprehensive view of the model's intricacies, we've made modifications to Section 6.2, specifically the fifth paragraph, to include a detailed description of Figure.6. Specifically, the MAE (Masked Auto-Encoder) represents a specialized auto-encoder network model, conceptualized for the purpose of reconstructing the original signal based on partial observations. As is consistent with all auto-encoders, our method incorporates an encoder that maps the observed signal to a semantic vector and a decoder that reconstructs the original signal from this semantic vector. A distinguishing feature of our approach, when compared to conventional auto-encoders, is our adoption of an asymmetrical design. This design mandates that the larger encoder works solely on the partial observed signals (excluding mask tokens), while a lightweight decoder reconstructs the full signal from the semantic vector coupled with mask tokens.
Q6: The authors shall have to give the specific parameters for the analyses in Figure.6 and the Transfer learning network model in Figure.7.
A6: Thank you for your comments regarding our phase preprocessing figures. Our detailed responses are as follows.
(1) For the model parameters in Figure.6 that you mentioned, we have provided a more comprehensive list and explanation of the parameters used in the revised manuscript, lines 495-497 and 508-515. Our MAE Encoder employs 8 Transformer Encoder Layers, while the MAE Decoder utilizes 5 Transformer Encoder Layers. In the masking layer, we chose a masking ratio of 40%, meaning that 40% of the unlabeled phase data would be randomly masked. However, the added learnable embedding header vectors will not be masked during this operation.
(2) Regarding the Transfer learning network model illustrated in Figure.7, we have enhanced the graphical content and provided a more in-depth description of the model's structure and the characteristics of each layer in Section 6.3, paragraph 2. The ViT transfer learning network first receives the semantic vectors output by the MAE Encoder. During computation, the ViT transfer learning network uses 5 Transformer Encoder Layers. Each Transformer Encoder Layer has an input feature dimension of 64, a multi-head attention count of 16, and a feed-forward network dimension of 2048. Layer normalization (LN) is employed internally.
Q7: The authors shall have to mention the utilized parameters for MSE Loss vs Epoch for Figure.12.
A7: Thank you for your valuable feedback on our manuscript. Pertaining to Figure.12, which illustrates the "MSE Loss vs Epoch", we have provided a comprehensive explanation about the related concepts in the revised version, specifically between lines 711-714. Moreover, we have attached the corresponding MSE calculation formula, as demonstrated in Equation (12).
Q8: The authors shall have to express the idea for more accuracy in that analyses.
A8: We appreciate your insightful suggestion. In the revised manuscript, specifically in Chapter 8, paragraphs 1-2, we have delved into a more detailed and profound analysis. By integrating experimental data with our neural network design rationale, we further substantiate our analysis and viewpoints. The empirical results indicate advancements of our approach even in high-cost data acquisition scenarios, as we can achieve more precise heart rate estimates with fewer labeled data in a shorter time span. In terms of the deep network architecture of our method, this study draws inspiration from current high-performing natural language processing network models, employing an innovative approach for heart rate estimation, which results in significantly enhanced accuracy.
Q9: The limitations and recommendation of the proposed work shall have to be discuss in the separate section before conclusion.
A9: Thank you for your valuable feedback. In the revised manuscript, we have added Section 8, titled "Discussion and Analysis," where we delve into the accomplishments and limitations of our research.
This section outlines some challenges addressed by our method, highlighting how it enhances the precision of non-contact heart rate estimation while reducing data collection costs and network complexity. However, there are certain limitations, such as the experiments being confined to subjects in a seated position, the radar's sensitivity to minor human movements, and the primary data source being healthy individuals. In future research, we will focus on enhancing the algorithm's robustness to subtle movements and consider collecting data from subjects with diverse health conditions to augment the model's adaptability and accuracy.
Comments on the Quality of English Language
Q1: The writing style shall have to be modified with Extensive editing of English language required.
A1: Thank you for your feedback. We have thoroughly revised the manuscript and sought the expertise of a professional English editing service to ensure clarity and precision.

Reviewer 3 Report
Major Comments:
1) Although the authors talked about self-attention mechanism in line 180, its advantage highlighted in lines 181-183 written as “While CNN and RNN architectures are usually fixed, MVN networks based on a self-attentive mechanism can better handle data of different sizes and scales, thus improving the model's generalization capability and making it easier to scale to larger image sizes and more complex tasks”. Not sure how this is related to self-attention. It should be elaborated.
2) The sentence in line 190 needs more explanation.
3) The claim in line 132 “ The introduction of neural networks solves the harmonic and DC offset problems” needs explanation.
Minor Comments:
1) The flowchart figure 2 should be centered.
2) Conclusion is small. Another paragraph can be added.
Author Response
Dear Editors and Reviewers:
Thank you for your letter and the reviewers’ comments on our manuscript entitled “MAE-based self-supervised pretraining algorithm for heart rate” (ID: sensors-2535326). These comments are valuable for improving our manuscript, as well as the important guiding significance to our research. We have answered the comments carefully and made corresponding corrections. We hope the revisions meet with approval. Revised portions are marked in red on our manuscript. The main corrections in the paper and the responses to the reviewers’ comments are as follows with reviewers’ comments in red.
Major Comments:
Q1: Although the authors talked about self-attention mechanism in line 180, its advantage highlighted in lines 181-183 written as “While CNN and RNN architectures are usually fixed, MVN networks based on a self-attentive mechanism can better handle data of different sizes and scales, thus improving the model's generalization capability and making it easier to scale to larger image sizes and more complex tasks”. Not sure how this is related to self-attention. It should be elaborated.
A1: Thank you for your insightful feedback regarding the self-attention mechanism discussed in our paper. In response to your query, we have made modifications and provided a more detailed explanation on the workings of the self-attention mechanism and its impact on model generalization and scalability in lines 246-256 of the revised manuscript.
The revised section explains the inherent limitations of CNN and RNN due to their fixed architectures: CNNs have a fixed filter size and RNNs process sequences in a predetermined order, suggesting their potential inadequacies in handling data of diverse sizes and shapes. In contrast, the self-attention mechanism dynamically allocates weights to each element in the input data, offering adaptability to varying data sizes and structures. Models based on the Transformer architecture, which employ self-attention, are also better equipped to prevent gradient vanishing or exploding issues, allowing for stacking more layers. As a result, MVN networks with self-attention mechanisms demonstrate enhanced generalization capabilities, and they more seamlessly scale to larger or more intricate tasks.
Q2: The sentence in line 190 needs more explanation.
A2: We appreciate your insightful suggestion. In reference to the sentence in line 190: "Therefore, compared with CNN for image data and RNN for temporal data, the MVN network exhibits superior performance in processing diverse data types," a more comprehensive discussion has been included in lines 261-271 of the revised manuscript.
The self-attention mechanism is proficient in concurrently processing all elements within a sequence and allocating distinct weights to each element. This mechanism enables the Transformer to take into account long-range dependencies in sequences, rather than being confined by the convolutional kernel size, as seen in CNNs, or relying on recursive processing typical of RNNs. Given that the MVN network integrates multi-head self-attention and multiple stacked layers, it possesses the capability to discern intricate patterns and relationships in extensive datasets. This positions it as having enhanced potential in the extraction and processing of radar data.
Q3: The claim in line 132 “ The introduction of neural networks solves the harmonic and DC offset problems” needs explanation.
A3: Thank you for your feedback regarding line 132. For clarity, we have revised the manuscript in lines 191-196. We elaborated that neural networks can effectively learn the nonlinear relationships in data, so they can still learn and correct for such DC offset issues. Furthermore, we detailed how neural networks avoid harmonic issues by directly learning from phase data. We hope these additions and clarifications make the manuscript's narrative clearer and more comprehensible.
Minor Comments:
Q1: The flowchart figure 2 should be centered.
A1: Thank you for your suggestion. We have adjusted Figure 2 to ensure the flowchart is centered and clearly displayed.
Q2: Conclusion is small. Another paragraph can be added.
A2: Thank you for your suggestion. In the revised manuscript, we have expanded Section 8 by incorporating a second paragraph. This new addition delves into the prospective focal points of our research, as outlined below.
(1) Comparing the proposed method with other non-contact solutions.
(2) Reconstructing the ECG and PPG of the examined individuals.
(3) Assessing if users have cardiovascular ailments or related conditions.

Round 2
Reviewer 1 Report
The authors gave fairly detailed answers to the comments and made corrections to the text of the article. I think the article can be published.